# An exploratory analysis of changes in work values among nurses before and after pregnancy

Yukari Hara[1][*], Aoi Nakagawa[2], Junko Omori[3]

1 Graduate School of Medicine, Tohoku University, Sendai, Miyagi, Japan, 2 Yamagata Prefecture Shonai General Branch Office Health Planning Division, Higashitagawa, Yamagata, Japan, 3 St. Luke's International University Graduate School of Nursing, Chuo-ku, Tokyo, Japan

☯ These authors contributed equally to this work.
* y.hara@tohoku.ac.jp

## Abstract

This study aimed to investigate changes in work values among Japanese nurses before and after pregnancy, their return to work while managing childcare responsibilities, and the multifaceted factors influencing these changes. A web-based survey of 199 female nurses assessed their work values before and after pregnancy, including retrospective questions and open-ended responses. Data were analyzed comprehensively using paired t-tests, generalized estimating equations, and inductive thematic analysis. Only prestige work values demonstrated a significant decrease after pregnancy, as revealed by paired t-tests. Generalized estimating equation analysis identified age at first child, social support from family and other sources, employment status, and educational background as the main factors influencing changes in work values. Qualitative findings indicated that this decline in prestige work values was due to a shift in nurses' awareness that "their life became centered on their children and family," alongside family roles and time constraints that limited career development. This study demonstrates that Japanese nurses experience significant changes in their work values during the transition from childbirth to work re-entry. These changes are intricately shaped by several factors, particularly individual life stage variables and social support from family and other sources. A nuanced understanding of these shifts in work values is essential for developing effective and individualized support systems to promote the retention and long-term career development of female nurses in Japan.

## Introduction

Basic human values are personal aspects and are highly stable after being formed during early childhood and adolescence [1,2]. However, previous research has shown that even after values are formed, they vary depending on the characteristics of each generation or age [2]. Approximately 7% of individuals undergo a significant

**Data availability statement:** All relevant data are within the manuscript and its Supporting Information files.

**Funding:** this research was funded by Japan Society for the Promotion of Science (JSPS) Grants-in-Aid for Scientific Research (KAKENHI) (grant number 22K17434) and TUMUG Support Program from Center for Diversity, Equity, and Inclusion, Tohoku University. The funders had no role in study design, data collection and analysis, decision to publish, or preparation of the manuscript.

**Competing interests:** The authors have declared that no competing interests exist.

shift in their values [3]. Changes in basic human values refer to changes in established value hierarchies and priorities [4]. These changes can be triggered by external events such as the global financial crisis [5] and the COVID-19 pandemic [6], as well as major life transitions such as becoming a parent [4].

In a professional context, reflecting these basic values, work values are defined as enduring beliefs about favorable work environments and desired outcomes [7]. Research has shown that work values are also susceptible to changes after significant life events, such as the onset of breast cancer [8] or the COVID-19 pandemic [9]. As motherhood leads to a shift in basic human values, with conservatism becoming more important than openness to change [4], it is hypothesized that similar shifts in work values occur. However, the specific nature of the shifts in work values due to parenthood and the multifaceted factors influencing these changes remain unclear.

This knowledge gap is particularly critical in nursing, where professional commitment and career continuity are paramount. Given the global nurse shortage and the fact that the majority of nurses are women [10], understanding these dynamics is urgent. This urgency is amplified in contexts like Japan, which faces significant challenges in gender equality, ranking 118th out of 146 countries on the Gender Gap Index [11]. In Japan, female labor force participation rates are low compared to other developed countries [12], and Japanese nurses often leave their jobs due to marriage, pregnancy, and childbirth [13]. In this context, work values are strongly linked to career continuity and affect turnover intentions [14], work engagement [15,16], and job satisfaction [17]. Therefore, understanding the changes in work values associated with child-rearing is essential for developing effective human resource retention strategies in an aging society.

Despite the importance of this issue, previous research lacks detailed insights into longitudinal changes in work values and the factors influencing such changes. For instance, while changes in work values in women who have experienced breast cancer are influenced by factors such as the stage of cancer, not being a manager, and the impact of cancer on daily life [8]. However, to the best of our knowledge, it is unclear the effects of nurses' personal attributes, the environment they live in, and whether their work values change because of becoming a parent. It is important to understand whether changes in nurses' work values are influenced by personal characteristics such as health status, the home environment where they raise their children, and the work environment after returning to work. Furthermore, research on basic values has shown that it takes several years for values to change [2]. For example, work values in women with breast cancer take two years to change [8]. Similarly, because changes in work values may not be fully recognized during parental leave, analyzing the trajectory from before pregnancy to after work re-entry can provide a comprehensive understanding of the change process.

To address the aforementioned knowledge gaps, this study aimed to clarify changes in work values from before pregnancy to after returning to work as a nurse. This approach is important for comprehensively understanding potential changes in work values that occur during parental leave and after returning to work. Specifically, it examines the changes in work values among Japanese nurses with children from

before pregnancy to work re-entry after childbirth and childcare, as well as the factors that influence these changes, by conducting primarily quantitative exploratory analysis and supplemented with qualitative insights.

## Methods

### Study design

The study employed a primarily quantitative cross-sectional observational design, using a web survey. To further interpret the quantitative findings, supplementary inductive thematic analysis of open-ended responses was incorporated. Retrospective survey questions were administered to clarify the changes in work values from before pregnancy to after returning to work as a nurse after childbirth and childcare. A longitudinal study with a clear chronological order and causal relationships would be ideal. However, longitudinal studies are difficult because of practical and methodological challenges [18]. For example, it is challenging to obtain and maintain a representative sample for a phenomenon wherein less than half of nurses return to work after pregnancy and childbirth, as in this study, owing to expected initial non-response and participant dropout. Therefore, this exploratory quantitative study with supplemental qualitative data was designed to guide future longitudinal research.

The validity of retrospective questionnaire surveys was reviewed by Thigpen [18] and Hipp et al. [19]. To mitigate potential recall bias, we carefully designed the retrospective questionnaire survey, referencing these studies. Specifically:

1. We established a clear anchor point for recall (just before the first pregnancy);

2. We used an established standardized scale (the Short version of the Nurses' Work Values Scale) to measure values at both time points;

3. We limited the sample to nurses who had returned to work, ensuring that the two time points were directly related to their current career experiences.

The supplementary use of inductive thematic analysis was incorporated to contextualize the extent of changes observed in the quantitative analysis and explore the underlying reasons and subjective experiences of value change, thereby enhancing the validity of the overall results. Additionally, we used the Strengthening the Reporting of Observational Studies in Epidemiology (STROBE statement) to improve study quality.

### Survey participants, sample size, and survey methods

The participants in this study were women who worked as nurses or licensed practical nurses and whose first child was between the ages of 0 and 6 (preschool age). To clarify changes in work values, an additional eligibility criterion was added: a requirement that more than one month had passed since their return to work after childcare leave. Using G*Power 3.1.9.6, we calculated the number of samples required for multiple regression analysis with an effect size of 0.15 (based on the small effect size in social psychology [20]), a significance level of 0.05, and a statistical power of 0.95 (to ensure high confidence in rejecting the null hypothesis [21]), which resulted in 199 cases. Therefore, we surveyed 200 individuals. An anonymous, self-administered online survey was conducted in Japan from July 26 to August 1, 2023. The questions created by the researchers were sent electronically to a research company conducting the academic research, which then created a webpage to post the questionnaire, screen participants, and collect data. The selection process involved conducting screening questions on the research company's online survey panel to ensure that only those who met all specified criteria (nurses/licensed practical nurses, first child aged 0–6 years, and returning to work for at least one month) could proceed to the main survey. Those who met the participation criteria proceeded to the survey screen. Participants provided explicit consent by checking the digital consent checkbox before answering the survey. A total of 200 participants completed the survey.

## Ethical considerations

The authors conducted this study in accordance with the ethical guidelines for medical and health research involving human participants in Japan and the 1995 Declaration of Helsinki, as revised in Edinburgh in 2000. Furthermore, this study was approved by the Ethics Committee of the Tohoku University Graduate School of Medicine, where the researchers belong (approval date: June 28, 2023, approval number: 2023-1-259). The survey was outsourced to an academic research company under the personal information protection regulations. At the beginning of the survey screen, an explanatory document was presented, explaining the study's purpose, objectives, significance, and methods, as well as information about maintaining anonymity and freedom to participate. Participants were informed that not participating would not pose disadvantages, that they could stop answering the survey at any time, and that they could only proceed with the questions if they had given explicit consent by checking the digital consent checkbox. The researchers received data from a survey company that did not include personal information, and the data files were strictly managed so that they would not be accessed by anyone other than the researchers. The results were statistically processed to ensure individual anonymity. Additionally, this study did not include minors. As all participants were required to be licensed nurses and parents, no minors were eligible for inclusion.

## Survey contents

Little is known about the factors that influence changes, except for age and generation. Based on prior evidence linking changes in work values among women who had experienced breast cancer to factors such as the stage of cancer, not being a manager, and the impact of cancer on daily life [8], we included variables related to the home environment for raising children, the work environment, and support from family and the workplace—all of which affect the balance between childcare and work [22–25]. The variables used in this study, including continuous and categorical variables, are described in detail in the Supporting Information (S1 Appendix).

**Participant characteristics, home, and work environments.** We collected data on participants' age, educational background, physical and emotional condition, health, time spent relaxing, time spent with their children, and financial situation. Family factors included marital status, family members living together, number of children, age of the first child, having someone to help with childcare, taking parental leave when their first child was born, childcare facility use, concerns about their children's development, raising their children as they desired, and having family members to provide care.

Regarding workplace factors, participants were asked about job type, years of nursing experience, workplace classification, years working in the current workplace, employment type, and working hours per week.

**Social support from family, friends, and significant others.** We used the Japanese version of the Multidimensional Social Support Perception Scale developed by Zimet et al. [26] and translated by Iwasa et al. [27]. This scale comprises three factors (12 items): family (4 items, e.g., "My family really helps me"), friends (4 items, e.g., "I have friends who I can share my joys and sorrows with"), and significant others (4 items, e.g., "I have someone who helps me when I'm in trouble"). Participants were asked to respond to each item on a 7-point scale, ranging from 1 (not at all) to 7 (very much). The sum of these scores was calculated as the social support score from family and others. A higher score indicates a stronger perception of social support. The descriptive statistics and Cronbach's alpha coefficients for each subscale and total scale for this study are shown in Table 1.

**Social support at work.** Items related to social support from superiors and colleagues were taken from the "social support" item of the Occupational Stress Simple Questionnaire [28]. Unlike conventional questionnaires that measure only stress reactions, this questionnaire also evaluates workplace stress simultaneously and can be used in various workplaces across different industries. Questions related to superiors and colleagues included three items: "How easy is it to consult with the following people?", "How reliable are the following people when you are in trouble?", and "How receptive are the following people when you consult with them about personal problems?" Responses were rated on a four-point scale ranging from 1 (not at all) to 4 (very much). The total score was calculated as the social support score at

**Table 1. Descriptive statistics of the scales, Cronbach's α coefficient, and paired t-tests using pre-pregnancy and during the survey.**

| | Scale | Minimum value | Maximum value | Mean | SD† | Cron-bach's α | p value‡ | Cohen's d | 95% confidence interval |
|---|---|---|---|---|---|---|---|---|---|
| Social support from family, friends, and significant others | | 13 | 84 | 64.77 | 12.39 | 0.95 | | | |
| | Family | 4 | 28 | 22.23 | 4.40 | 0.95 | | | |
| | Friends | 4 | 28 | 20.47 | 5.12 | 0.93 | | | |
| | Significant others | 5 | 28 | 22.07 | 4.21 | 0.91 | | | |
| Workplace social support | | 6 | 24 | 14.94 | 3.95 | 0.90 | | | |
| | Supervisor | 3 | 12 | 7.06 | 2.21 | 0.91 | | | |
| | Colleagues | 3 | 12 | 7.89 | 2.15 | 0.88 | | | |
| Intrinsic work values | Before the first pregnancy | 3 | 15 | 9.94 | 2.94 | 0.95 | 0.07 | 0.13 | [-0.03, 0.64] |
| | During the survey | 3 | 15 | 9.64 | 2.80 | 0.93 | | | |
| | Amount of change | −8 | 9 | −0.31 | 2.40 | | | | |
| Extrinsic work values | Before the first pregnancy | 3 | 15 | 10.62 | 2.83 | 0.86 | 0.75 | 0.02 | [-0.29, 0.40] |
| | During the survey | 3 | 15 | 10.57 | 2.58 | 0.78 | | | |
| | Amount of change | −8 | 8 | −0.06 | 2.48 | | | | |
| Social work values | Before the first pregnancy | 3 | 15 | 10.02 | 2.88 | 0.93 | 0.44 | 0.06 | [-0.19, 0.43] |
| | During the survey | 3 | 15 | 9.90 | 2.96 | 0.93 | | | |
| | Amount of change | −7 | 9 | −0.12 | 2.19 | | | | |
| Prestige work values | Before the first pregnancy | 3 | 15 | 9.38 | 3.07 | 0.94 | p < 0.01 | 0.22 | [0.17, 0.79] |
| | During the survey | 3 | 15 | 8.90 | 3.06 | 0.92 | | | |
| | Amount of change | −7 | 10 | −0.48 | 2.21 | | | | |

†SD: standard deviation, ‡Paired t-test.

work, with higher scores indicating a stronger perception of social support. Table 1 shows the descriptive statistics for this study and Cronbach's alpha coefficients for each subscale (supervisors and colleagues) and the total scale.

**Nurses' work values.** To assess nurses' work values, we used the "Short version of Nurses' Work Values Scale" developed by Hara et al. [29]. This established and validated scale is consistent with the theoretical foundations derived from Ros et al. [30] and consists of 12 items across four factors: three items each for intrinsic, extrinsic, social, and prestige work values. Intrinsic work values are sub-factors that represent autonomy in work and the pursuit of personal growth, while extrinsic work values are sub-factors that represent dimensions such as income gained from work, job security, and occupational stability. Social work values are sub-factors that represent contributions to society through work and having a positive relationship with society while working, while prestige work values are sub-factors that represent social status and recognition gained through work, as well as influence over others [30]. Examples of scale items are: "Improving nursing practice skills" and "Contributing to society by working as a nurse." Items were rated on a 5-point Likert scale, ranging from 1 to 5 (1 = not important at all; 2, not very important; 3, somewhat important; 4, quite important, and 5 = very important), with higher scores indicating that nurses placed more importance on the value.

Participants answered based on their situation during the survey in July-August 2023. They were presented with the same scale once more and were asked to choose the answer that applied to their situation before becoming pregnant with their first child. Specifically, when presented with the work values scale for the second time, participants were asked the following: "Reflect on the time before you became pregnant with your first child. How important were each of the following items to you at the time? Please choose the answer that best applies to each question." The descriptive statistics of this study and Cronbach's alpha coefficients for the four factors at two time points are shown in Table 1.

**Open-ended questions regarding value changes.** At the end of the online survey, we provided a free-form comment section where participants were asked to answer the following question: Have you ever felt that your values have changed

since you became pregnant with your first child? If so, what values do you think have changed and how? Please freely write about these points.

### Data analysis

**Analysis of quantitative variables.** First, descriptive statistics and reliability coefficients were calculated for individual factors, family factors, workplace factors, and the nurses' work values scale. To quantify the changes in the work values of nurses from before their first pregnancy to the time of the survey, descriptive statistics were calculated for the work values during the survey in July and August 2023 and before their first pregnancy. A paired t-test was then performed to examine whether the values had changed. To indicate the magnitude and precision of the change, we reported the effect size (Cohen's d) and 95% confidence intervals along with the results of the t-tests. The changes were calculated as the work value during the survey minus that before pregnancy. A positive change indicated that work values were more important during the survey than before pregnancy, whereas a negative change indicated that work values were less important during the survey than before pregnancy. The distribution of the changes was plotted to confirm the results.

Pearson product-moment correlation coefficients were calculated between the scale scores and the continuous variables used in this study. The correlation coefficient between the work values scores at two points in time was used to evaluate the stability of the relative positions of variables for individuals over time as rank-order stability [2,3]. Statistical analyses up to this point were performed using IBM SPSS ver.26.0 for Windows (IBM Corporation, Armonk, NY, USA). All tests were two-sided, at the 5% significance level. There were no missing values for any of the variables used in the analysis.

Finally, we used generalized estimating equations (GEE) to investigate associated factors, including potential confounders, with changes in nurses' four work values as the dependent variable. GEE is an extension of the generalized linear model and is suitable for data structures with correlations between observations, such as repeated measurements and clustered data [31]. As the retrospective questionnaire survey in this study was expected to have intrapersonal correlations between the change in the four work value factors before pregnancy and at the time of the survey, we conducted a regression analysis using GEE with changes as the dependent variable.

Independent variables included age, educational background, marital status, family members living together, number of children, age of first child, presence of a helper in childcare, presence of childcare leave, use of childcare facilities, presence of family caregivers, physical and emotional status, health status, presence of concerns about the child's development, spending relaxed time with children, economic situation, ideal child-rearing, workplace classification, years of work at the current workplace, employment type, presence of night shifts, working hours, multidimensional perceived social support scale, and social support in the workplace. All continuous variables were standardized to z-scores before analysis to allow estimates to be interpreted as standardized regression coefficients (effect sizes) [32].

For categorical variables, Cohen's d was calculated as the effect size, and the between-group difference in least-squares means (LSMEANS) obtained by GEE maximum likelihood estimation was divided by the standard deviation of the dependent variable. Furthermore, as a GEE model diagnostic, two working correlation structures, independence and exchangeable, were compared using the quasi-independence model quasi-likelihood criterion (QIC). To ensure transparency, we also report the unadjusted version of QIC (QICu) based on model-based covariance. For two-timepoint data, AR (1) and unstructured were excluded from the comparison because they are mathematically equivalent to the exchangeable structure. Since the QIC was the same for both structures, the exchangeable structure was used for the final model. Statistical analysis was performed using PROC GENMOD in SAS version 9.4 (SAS Institute Inc., Cary, NC, USA), with a significance level of two-sided $p < 0.05$.

**Inductive thematic analysis of descriptive statistics.** The open-ended responses were analyzed using inductive thematic analysis to gain a nuanced understanding of the subjective experiences underlying quantitative changes in work values. Specifically, the analysis was conducted using the following steps. First, the open-ended responses regarding changes in values were read multiple times to identify common meanings and patterns throughout the data, as well as points emphasized by participants. We aimed to capture participants' words as faithfully as possible, avoiding

over-interpretation by the researchers. Similar data were inductively grouped to identify abstract subcategories and categories. In this process, a categorical structure was constructed that could comprehensively explain the entire dataset, taking into account the relationships and hierarchies between codes. To ensure the trustworthiness and rigor of the findings, the initial analysis was conducted by one researcher, followed by a peer debriefing process where two other researchers independently reviewed the coding and category development. Any discrepancies in interpretation were discussed until consensus was reached, ensuring that the final categories accurately reflected the participants' perspectives. No qualitative data analysis software was used in the analysis process. Finally, the extracted subcategories and categories were reviewed to complement the interpretation of the results obtained from the quantitative analysis (a narrative synthesis strategy) [33]. For data tracking and integration purposes, each qualitative response was assigned a de-identified participant ID. This ID serves to link data points and does not identify individuals.

## Results

### Participant characteristics

Responses were obtained from 200 participants. One person who answered "Currently taking maternity leave/childcare leave" was excluded; thus, 199 responses were analyzed (valid response rate: 99.50%). Participants' personal attributes are listed in Table 2.

### Changes in work values from before the first pregnancy to the time of the survey

Paired t-tests were conducted using the values before the first pregnancy and during the survey. The results were as follows: intrinsic work values, $t(198) = 1.80$, $p = 0.07$, 95% CI [−0.03, 0.64], Cohen's $d = 0.13$; extrinsic work values, $t(198) = 0.32$, $p = 0.75$, 95% CI [−0.29, 0.40], Cohen's $d = 0.02$; social work values, $t(198) = 0.78$, $p = 0.44$, 95% CI [−0.19, 0.43], Cohen's $d = 0.06$; prestige work values, $t(198) = 3.05$, $p = 0.003$, 95% CI [0.17, 0.79], Cohen's $d = 0.22$. These results indicate that of the four work value scales, only prestige work values showed a significant decrease after pregnancy (Table 1). This finding suggests that participants' values and motivational priorities may have changed. However, because the effect size was small, this change may represent a gradual and subtle shift in priorities rather than a dramatic shift in values.

The scores of the four work value factors before the first pregnancy were subtracted from the scores during the survey, and the amount of change was plotted (Fig 1). In total, the importance of intrinsic, extrinsic, social, and prestige values did not change in 81, 67, 83, and 85 participants, respectively. In terms of intrinsic, extrinsic, social, and prestige values, 44, 58, 52, and 40 people, respectively, showed a change of one or more (i.e., increased importance of values); while 74, 74, 64, and 74 people, respectively, showed a change of less than one (i.e., decreased importance of values). Overall, more people tended to have unchanged values, but more people displayed decreased rather than increased importance of work values. Looking at the four factors, the number of people whose extrinsic work values changed, and the number of people whose importance increased was the largest among the four factors. Conversely, the number of people whose importance decreased was the smallest among the four factors for social work values.

### Correlation of each variable

The correlation coefficient results for each variable are shown in Table 3. The correlation coefficients between pre-pregnancy and the time of the survey for intrinsic, extrinsic, social, and prestige work values were 0.650, 0.584, 0.719, and 0.740, respectively, with prestige work values showing the highest rank-order stability and extrinsic work values showing the lowest.

### Generalized estimating equations

GEE models were performed to examine factors associated with the changes in each of the four work values among nurses. Because all continuous predictors were standardized prior to analysis, the parameter estimates can be directly interpreted as

**Table 2. Demographic characteristics of the participants (N = 199).**

| Items | N(%)/Mean±SD |
|---|---|
| Age | 33.80 ± 5.06 |
| Educational background | |
| University/Graduate School | 78(39.20) |
| Other | 121(60.80) |
| Physical and emotional status | |
| In good condition both physically and mentally | 110(55.28) |
| Not in good condition both physically and mentally | 89(44.72) |
| Health status | |
| Good | 94(47.24) |
| Average | 69(34.67) |
| Bad | 36(18.09) |
| Have time to spend with their children in a relaxed atmosphere | |
| Yes | 114(57.29) |
| No | 85(42.71) |
| Financial situation | |
| Comfortable | 67(33.67) |
| Not comfortable | 132(66.33) |
| Marital status | |
| Married | 185(92.96) |
| Unmarried, Other | 14(7.04) |
| Family member living in a | |
| Single-mother household | 10(5.03) |
| Three-generation household | 14(7.04) |
| Other | 175(87.94) |
| Number of children | |
| 1 | 114(57.29) |
| 2 | 77(38.69) |
| 3 | 8(4.02) |
| Age of first child (years) | |
| First child | 3.83 ± 1.74 |
| Childcare helper | |
| Yes | 188(94.47) |
| No | 11(5.53) |
| Childcare leave taken | |
| Yes | 175(87.94) |
| No | 24(12.06) |
| Childcare facility use | |
| Yes | 157(78.89) |
| No | 42(21.11) |
| Family member providing care | |
| Yes | 5(2.51) |
| No | 194(97.49) |
| Worried about child's development | |
| Yes | 50(25.13) |
| No | 149(74.87) |

*(Continued)*

**Table 2.** (Continued)

| Items | N(%)/Mean±SD |
|---|---|
| Ideal parenting | |
| Yes | 90(45.23) |
| No | 109(54.77) |
| Years of experience in nursing | 10.92±4.74 |
| Workplace classification | |
| Hospital | 157(78.89) |
| Non-hospital | 42(21.11) |
| Number of years working in the current job | 5.45±3.83 |
| Employment status | |
| Regular full-time | 117(58.79) |
| Regular part-time | 63(31.66) |
| Non-regular | 19(9.55) |
| Working hours per a week | |
| 1–45 hours | 184(92.46) |
| 46 hours or more | 15(7.54) |

SD: standard deviation.

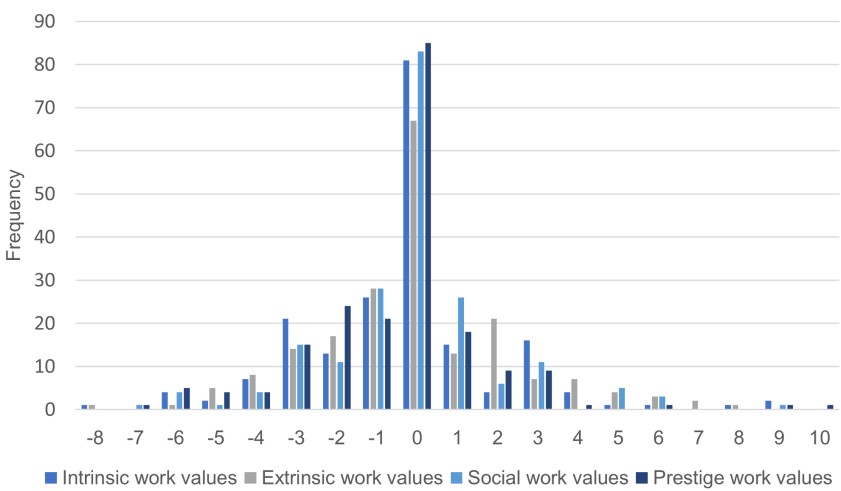

**Fig 1. Differences in the total scores of intrinsic, extrinsic, social, and prestige work values before pregnancy and during the study.**

standardized regression coefficients. For categorical predictor variables, effect sizes—Cohen's d—were calculated based on the estimated LSMEANS difference obtained from the GEE model. Across all four GEE models, the exchangeable working correlation structure was adopted because it demonstrated identical model fit to the independence structure based on QIC and QICu. For changes in intrinsic work values, both structures yielded the same QIC (218.12) and QICu (224). Similarly, for extrinsic work values, QIC was 217.28 and QICu was 224 for both structures. The same pattern was observed for changes in social work values (QIC = 218.22; QICu = 224) and prestige work values (QIC = 217.33; QICu = 224).

**Factors related to changes in intrinsic work values.** Table 4 presents the GEE analysis results, with the changes in the four factors of nurses' work values from before the first pregnancy to the time of the survey as the dependent variable.

**Table 3. Correlation matrix.**

| | 1 | 2 | 3 | 4 | 5 | 6 | 7 | 8 | 9 | 10 | 11 | 12 | 13 | 14 | 15 | 16 | 17 |
|---|---|---|---|---|---|---|---|---|---|---|---|---|---|---|---|---|---|
| 1 Age | — | | | | | | | | | | | | | | | | |
| 2 Number of children | 0.069 | — | | | | | | | | | | | | | | | |
| 3 Age of first child | 0.369*** | 0.452*** | — | | | | | | | | | | | | | | |
| 4 Number of years working in the current job | 0.174* | 0.042 | 0.080 | — | | | | | | | | | | | | | |
| 5 Social support from family, friends, and significant others | -0.220** | 0.070 | -0.108 | -0.036 | — | | | | | | | | | | | | |
| 6 Workplace social support | -0.015 | 0.000 | 0.119 | -0.054 | 0.280*** | — | | | | | | | | | | | |
| 7 Intrinsic work values during the survey | 0.094 | -0.047 | 0.000 | -0.010 | 0.093 | 0.330*** | — | | | | | | | | | | |
| 8 Extrinsic work values during the survey | 0.068 | -0.068 | -0.109 | 0.046 | 0.139 | 0.231** | 0.517*** | — | | | | | | | | | |
| 9 Social work values during the survey | 0.079 | 0.003 | -0.007 | -0.041 | 0.193** | 0.338*** | 0.763*** | 0.544*** | — | | | | | | | | |
| 10 Prestige work values during the survey | 0.040 | -0.043 | -0.110 | -0.031 | 0.180* | 0.266*** | 0.661*** | 0.489*** | 0.769*** | — | | | | | | | |
| 11 Intrinsic work values before first pregnancy | 0.035 | -0.064 | 0.055 | -0.038 | 0.317*** | 0.423*** | 0.650*** | 0.355*** | 0.664*** | 0.549*** | — | | | | | | |
| 12 Extrinsic work values before first pregnancy | -0.038 | -0.010 | 0.021 | -0.070 | 0.267*** | 0.337*** | 0.274*** | 0.584*** | 0.358*** | 0.368*** | 0.436*** | — | | | | | |
| 13 Social work values before first pregnancy | 0.024 | -0.073 | 0.004 | -0.065 | 0.316*** | 0.411*** | 0.528*** | 0.415*** | 0.719*** | 0.575*** | 0.797*** | 0.491*** | — | | | | |
| 14 Prestige work values before first pregnancy | 0.005 | -0.105 | -0.071 | -0.041 | 0.245*** | 0.295*** | 0.466*** | 0.341*** | 0.576*** | 0.740*** | 0.680*** | 0.469*** | 0.763*** | — | | | |
| 15 Amount of change in intrinsic work values | -0.074 | -0.016 | 0.064 | -0.032 | 0.256*** | 0.090 | -0.464*** | -0.218** | -0.159* | -0.168* | 0.371*** | 0.174* | 0.283*** | 0.223** | — | | |

*(Continued)*

**Table 3.** (Continued)

| | 1 | 2 | 3 | 4 | 5 | 6 | 7 | 8 | 9 | 10 | 11 | 12 | 13 | 14 | 15 | 16 | 17 |
|---|---|---|---|---|---|---|---|---|---|---|---|---|---|---|---|---|---|
| 16 Amount of change in extrinsic work values | −0.117 | 0.068 | 0.147* | −0.125 | 0.119 | 0.086 | −0.306*** | −0.536*** | −0.249*** | −0.177* | 0.048 | 0.373*** | 0.037 | 0.098 | 0.430*** | — | |
| 17 Amount of change in social work values | −0.072 | −0.103 | 0.015 | −0.033 | 0.173* | 0.111 | −0.290*** | −0.155* | −0.343*** | −0.235** | 0.204** | 0.194** | 0.406*** | 0.274*** | 0.592*** | 0.378*** | — |
| 18 Amount of change in prestige work values | −0.048 | −0.086 | 0.053 | −0.014 | 0.089 | 0.040 | −0.272*** | −0.206** | −0.270*** | −0.362*** | 0.181* | 0.140* | 0.259*** | 0.359*** | 0.543*** | 0.380*** | 0.705*** |

* $p < .05$, **$p < .01$, ***$p < .001$.

Pearson's product–moment correlation coefficients.

**Table 4. Results of generalized estimating equations using the four subdomains of work values as dependent variables (N = 199).**

| | Changes in intrinsic work values | | | | | | Changes in extrinsic work values | | | | | |
| | 95% confidence interval | | | | | | 95% confidence interval | | | | | |
| | β | Cohen's d | Standard error | Lower limit | Upper limit | p | β | Cohen's d | Standard error | Lower limit | Upper limit | p |
|---|---|---|---|---|---|---|---|---|---|---|---|---|
| Age | −.129 | | .082 | −.289 | .032 | .115 | **−.169** | | **.070** | **−.306** | **−.031** | **.016*** |
| Educational background (University/Graduate School = 1) | .193 | .193 | .138 | −.076 | .463 | .160 | .266 | .266 | .139 | −.005 | .538 | .055 |
| Marital status (Married = 1) | .407 | .407 | .354 | −.287 | 1.102 | .250 | .299 | .299 | .295 | −.278 | .877 | .310 |
| Family member living in a (3 generation household = 1) | .164 | .163 | .302 | −.428 | .755 | .588 | .379 | .380 | .240 | −.091 | .850 | .114 |
| Number of children | −.078 | | .074 | −.224 | .068 | .294 | −.024 | | .071 | −.163 | .115 | .736 |
| Age of first child | **.219** | | **.077** | **.068** | **.370** | **.005**** | **.277** | | **.081** | **.119** | **.434** | **.001***** |
| Childcare helper (Yes = 1) | −.404 | −.403 | .323 | −1.036 | .229 | .211 | −.136 | −.136 | .321 | −.765 | .493 | .671 |
| Childcare leave taken (Yes = 1) | .184 | .183 | .233 | −.273 | .640 | .431 | −.482 | −.483 | .253 | −.978 | .013 | .056 |
| Childcare facility use (Yes = 1) | .122 | .122 | .134 | −.141 | .386 | .363 | .052 | .052 | .152 | −.246 | .350 | .731 |
| Family member providing care (Yes = 1) | −.080 | −.080 | .284 | −.636 | .476 | .778 | −.621 | −.621 | .265 | −1.141 | −.101 | .019 |
| Physical and emotional status (In good condition both physically and mentally = 1) | −.143 | −.143 | .163 | −.462 | .176 | .379 | −.032 | −.032 | .158 | −.341 | .278 | .842 |
| Health status (Good = 1) | −.211 | −.210 | .146 | −.497 | .075 | .148 | .015 | .015 | .148 | −.275 | .306 | .918 |
| Worried about your child's development (Yes = 1) | −.202 | −.202 | .167 | −.529 | .124 | .225 | .114 | .114 | .160 | −.200 | .428 | .477 |
| Have time to spend with their children in a relaxed atmosphere (Yes = 1) | −.121 | −.120 | .141 | −.397 | .156 | .393 | −.107 | −.107 | .148 | −.398 | .183 | .469 |
| Financial situation (Comfortable = 1) | −.228 | −.228 | .132 | −.487 | .031 | .084 | −.044 | −.044 | .145 | −.327 | .240 | .763 |
| Ideal parenting (Yes = 1) | .058 | .058 | .137 | −.210 | .326 | .671 | −.059 | −.059 | .171 | −.395 | .277 | .732 |
| Workplace classification (Hospital = 1) | −.234 | −.234 | .161 | −.550 | .082 | .147 | .083 | .083 | .169 | −.249 | .415 | .623 |
| Number of years working in the current job | −.018 | | .075 | −.165 | .130 | .816 | −.098 | | .071 | −.238 | .042 | .168 |
| Employment status (Regular full-time = 1) | .242 | .242 | .149 | −.050 | .534 | .104 | **.384** | **.384** | **.148** | **.094** | **.674** | **.009**** |
| Night shift (None = 1) | .048 | .048 | .133 | −.212 | .308 | .718 | .150 | .150 | .130 | −.105 | .404 | .249 |
| Working hours per a week (1–45 hours = 1) | .309 | .309 | .243 | −.166 | .785 | .202 | .137 | .137 | .262 | −.376 | .650 | .601 |
| Social support from family, friends, and significant others | **.271** | | **.081** | **.113** | **.429** | **.001***** | .061 | | .081 | −.098 | .219 | .454 |
| Workplace social support | −.138 | | .076 | −.287 | .011 | .069 | −.070 | | .067 | −.201 | .062 | .299 |
| Intrinsic/extrinsic/social/prestige work values during the survey | **.381** | | **.079** | **.226** | **.537** | **<.0001***** | **.400** | | **.074** | **.255** | **.545** | **<.0001***** |

¹β values represent standardized regression coefficients because all continuous predictors were z-standardized prior to analysis.

²Cohen's d values represent adjusted effect sizes for categorical variables, calculated from the adjusted LSMEANS differences obtained from the GEE model and divided by the standard deviation of the standardized outcome (SD = 1).

³All GEE models used an exchangeable working correlation structure, selected based on model fit equivalence (QIC) across independence and exchangeable structures.

⁴* p < .05, **p < .01, ***p < .001.

Older ages of the first child were associated with greater increases in intrinsic work values (β = 0.219, SE = 0.077, 95% CI [0.068, 0.370], p = .005). Higher social support from family, friends, and significant others (MSPSS) was also positively associated with changes in intrinsic work values (β = 0.271, SE = 0.081, 95% CI [0.113, 0.429], p = .001). Baseline work-value scores strongly predicted subsequent changes (β = 0.381, SE = 0.079, p < .0001). All other predictors, including demographic and employment-related variables, showed negligible standardized effects (|β| < 0.23; all p > .05), with small effect sizes for categorical predictors (e.g., marital status: d = 0.407).

| Changes in social work values | | | | | | Changes in prestige work values | | | | | |
| 95% confidence interval | | | | | | 95% confidence interval | | | | | |
| β | Cohen's d | Standard error | Lower limit | Upper limit | p | β | Cohen's d | Standard error | Lower limit | Upper limit | p |
|---|---|---|---|---|---|---|---|---|---|---|---|
| −.107 | | .084 | −.270 | .057 | .203 | −.124 | | .088 | −.297 | .049 | .159 |
| **.330** | **.330** | **.148** | **.040** | **.619** | **.026*** | .174 | .174 | .160 | −.140 | .487 | .278 |
| .368 | .368 | .289 | −.198 | .934 | .202 | −.114 | −.114 | .208 | −.522 | .294 | .584 |
| .293 | .294 | .268 | −.232 | .818 | .273 | .091 | .091 | .207 | −.314 | .497 | .658 |
| −.141 | | .073 | −.283 | .002 | .054 | −.115 | | .073 | −.257 | .027 | .113 |
| **.229** | | **.083** | **.066** | **.391** | **.006**** | **.239** | | **.091** | **.061** | **.417** | **.009**** |
| −.381 | −.381 | .285 | −.938 | .177 | .181 | .044 | .044 | .226 | −.399 | .487 | .846 |
| .058 | .058 | .229 | −.391 | .507 | .801 | .119 | .119 | .247 | −.364 | .602 | .630 |
| −.225 | −.225 | .143 | −.506 | .056 | .117 | −.082 | −.082 | .158 | −.392 | .228 | .604 |
| −.278 | −.278 | .230 | −.729 | .173 | .227 | −.014 | −.014 | .352 | −.703 | .675 | .968 |
| −.110 | −.110 | .169 | −.442 | .222 | .516 | −.085 | −.085 | .192 | −.461 | .291 | .656 |
| −.143 | −.143 | .147 | −.431 | .145 | .331 | −.215 | −.215 | .163 | −.535 | .105 | .187 |
| −.216 | −.216 | .170 | −.550 | .117 | .204 | −.056 | −.056 | .160 | −.369 | .257 | .727 |
| −.028 | −.028 | .147 | −.317 | .261 | .848 | −.036 | −.036 | .146 | −.321 | .250 | .806 |
| −.259 | −.259 | .151 | −.555 | .038 | .087 | −.091 | −.091 | .170 | −.423 | .242 | .592 |
| .064 | .064 | .155 | −.239 | .368 | .679 | .109 | .109 | .158 | −.201 | .419 | .490 |
| −.099 | −.099 | .168 | −.428 | .231 | .558 | −.156 | −.156 | .182 | −.511 | .200 | .392 |
| .007 | | .080 | −.149 | .164 | .928 | .007 | | .080 | −.150 | .164 | .930 |
| .092 | .092 | .146 | −.194 | .378 | .530 | .060 | .060 | .150 | −.234 | .355 | .689 |
| −.112 | −.112 | .139 | −.385 | .160 | .420 | −.211 | −.210 | .141 | −.487 | .066 | .135 |
| .047 | .047 | .235 | −.414 | .508 | .842 | .405 | .404 | .259 | −.103 | .912 | .118 |
| .138 | | .071 | −.001 | .276 | .051 | .076 | | .079 | −.079 | .231 | .338 |
| −.088 | | .074 | −.234 | .058 | .238 | −.122 | | .072 | −.263 | .020 | .091 |
| **.427** | | **.087** | **.256** | **.598** | **<.0001***** | **.405** | | **.097** | **.216** | **.594** | **<.0001***** |

**Factors related to changes in extrinsic work values.** Age showed a small but statistically significant negative association with changes in extrinsic work values (β = −0.169, SE = 0.070, 95% CI [−0.306, −0.031], p = .016), indicating that younger nurses experienced slightly larger increases in extrinsic work values over time. Age of the first child was again positively associated with increases in extrinsic work values (β = 0.277, SE = 0.081, 95% CI [0.119, 0.434], p = .001). Employment status (regular full-time) demonstrated a positive effect (β = 0.384, SE = 0.148, 95% CI [0.094, 0.674], p = .009). Baseline work-value levels were strongly predictive of extrinsic change (β = 0.400, SE = 0.074, p < .0001). All remaining variables were nonsignificant with small standardized or categorical effect sizes.

**Factors related to changes in social work values.** Educational background was significantly associated with changes in social work values, with university/graduate-school-educated nurses showing greater increases (β = 0.330, SE = 0.148, 95% CI [0.040, 0.619], p = .026). Age of the first child also remained a consistent positive predictor (β = 0.229, SE = 0.083, 95% CI [0.066, 0.391], p = .006). Baseline work-value scores again strongly predicted increases in social work values (β = 0.427, SE = 0.087, p < .0001). Other predictors—including health, childcare support, or employment-related variables—showed small or nonsignificant effects (e.g., childcare leave taken: d = 0.058; family member caregiver: d = −0.278).

**Factors related to changes in prestige work values.** Age of the first child showed a positive association with changes in prestige work values (β = 0.239, SE = 0.091, 95% CI [0.061, 0.417], p = .009). Baseline work-value levels were again the strongest predictor (β = 0.405, SE = 0.097, p < .0001). All other predictors exhibited small or nonsignificant effects (all p > .05), with minimal effect sizes across categorical variables (|d| < 0.21).

**Summary of GEE results for the four work-value subdomains.** Across the four subdomains of work values, several consistent patterns emerged. The age of the first child was the most stable predictor, showing significant positive associations with increases in intrinsic, extrinsic, social, and prestige work values. Baseline work-value levels also demonstrated strong and consistent effects across all domains, indicating substantial temporal stability and carryover effects.

Perceived social support from family, friends, and significant others was associated with increases in intrinsic work values, although this relationship was not observed in the other domains. Educational background was a significant predictor only for social work values, where nurses with higher educational attainment showed larger increases. For extrinsic work values, younger age and regular full-time employment were associated with greater positive changes.

Most demographic, childcare-related, and workplace characteristics—such as marital status, family structure, childcare arrangements, health indicators, night-shift work, and working hours—demonstrated small or negligible effects, with Cohen's d values consistently indicating minimal impact across domains. Overall, the findings highlight that individual life-stage factors, particularly child-related variables and initial levels of work values, play a more prominent role in shaping changes in work values than structural or employment-related factors.

### Results of inductive thematic analysis

Of the 199 participants, 76 did not mention any changes in values in their freewriting descriptions. In total, 123 people wrote about changes in values. Inductive thematic analysis of the contents of the descriptions extracted three categories and eleven subcategories. A list of categories, subcategories, and examples of the included data are presented in Table 5.

## Discussion

This study explored changes in work values among Japanese nurses from before pregnancy to after childbirth and work re-entry, and the factors influencing them, using a primarily quantitative design, with supplemental qualitative insights. The quantitative analysis revealed a decline in prestige work values. Furthermore, analysis using GEE showed that factors such as the age of the first child, social support from family, friends, and significant others, and employment status influenced changes in these values. This indicates that as nurses experience pregnancy, childbirth, and childcare, their work values change in multiple ways due to various factors.

### Changes in work values

Regarding the specific transformations observed, a notable quantitative finding from this study was that the prestige work values of nurses decreased after returning to work compared to their pre-pregnancy scores (Table 1). When the changes calculated from the scores on the four work values factors before the first pregnancy and during the study period were plotted at the individual level (Fig 1), a trend toward stability was observed, with the majority of participants showing no

**Table 5. List of categories, subcategories, and examples of the included data.**

| Category | Subcategory | Examples of the included data | ID |
|---|---|---|---|
| My life became centered on my children and family | Children and family come first. | "Raising children took priority over work." | 5 |
| | | "My children came first, and I started to put work on the back burner." | 22 |
| | | "I had always prioritized work, but after having children, I was surprised to find that my priorities naturally changed." | 63 |
| | My sense of money changed. | "I want to earn as much as possible and help my family." | 168 |
| | | "I just want to live a stable life." | 58 |
| | | "It became more important to receive money, because my family is important, I don't work too hard." | 23 |
| Family and work must be balanced. | Balancing child-rearing and work is hard. | "I learned from experience that balancing child-rearing and work is very hard." | 12 |
| | | "Balancing work and child-rearing is hard." | 36 |
| | | "Balancing child-rearing and work is harder than I thought." | 110 |
| | Emphasis on work-life balance | "I started to think about work-life balance." | 61 |
| | | "I started to reflect on my family. I started thinking about work-life balance." | 69 |
| | | "I began to think that there is a great need to pursue work-life balance." | 196 |
| My thoughts and actions have changed | I work hard for my family | "I have to work hard for my family." | 3 |
| | | "I feel like I want to work hard for my children." | 55 |
| | | "Work is important, but I realized that my family is the most important thing. My children motivate me to work hard." | 65 |
| | My views on life and death have changed. | "That humans are born and alive is a miracle." | 166 |
| | | "My views on life and death have changed." | 193 |
| | My thoughts and actions have changed through the way I interact with others. | "I now often act while thinking about how others would feel if they were in my position." | 6 |
| | | "My range of thinking has broadened, and I am now able to compromise to a certain extent and forgive others." | 96 |
| | | "I have become more considerate." | 117 |
| | | "I have started to consider my child's and other people's positions when I say things." | 154 |
| | My thoughts and actions have changed through raising my child | "I realized that in raising children, it is important to be able to accept the extent to which you can be selfish." | 32 |
| | | "I used to be bad with children and used to hate children running around and shouting. I was also irritated with parents who didn't try to stop them, but I realized that each family has their own circumstances, and I was able to suppress my irritation. I began to respect people who are raising children around me." | 57 |
| | | "I realized the strength of a mother who lives her life while pouring love into her child." | 92 |
| | My thoughts and actions at work have changed. | "Since my child was born, I have been able to think more carefully and empathize with things by imagining what would happen to my family if what is happening to the patient in front of me were happening to my family." | 116 |
| | | "Because both mother and child nearly died during childbirth, death feels more real to me. I am more likely to empathize with patients and their families." | 142 |
| | | "In pediatric nursing, I have come to understand the feelings of parents and have more time for myself." | 192 |
| | My thoughts and actions at work have changed (negative). | "I felt that putting my children first meant that I would work less, which would affect my career. It takes time to build a career with reduced hours." | 107 |

*(Continued)*

**Table 5.** (Continued)

| Category | Subcategory | Examples of the included data | ID |
|---|---|---|---|
| | | *"I was no longer able to work as much, so my salary decreased, and I began to feel that I was not needed."* | 100 |
| | I started to value my life | *"I started to want to value my life."* | 14 |
| | | *"I started to live each day to the fullest."* | 17 |
| | | *"Time is limited, so I started to want to cherish each and every day."* | 191 |

changes in importance for intrinsic, extrinsic, social, and status values (81, 67, 83, and 85, respectively). However, the importance of values decreased more frequently than increased (e.g., prestige work values increased in 40 participants compared to a decrease in 74 participants). Prestige work values—which refer to a tendency to emphasize social status, promotion, and influence over others—declined and their distribution supported previous research findings that major life events alter the hierarchy of values [3,4]. This decline in prestige work values strongly integrated with findings from qualitative analysis of open-ended responses. The category extracted in the qualitative analysis, "My life became centered on my children and family," and particularly its subcategories, "Children and family come first," and "My thoughts and actions at work have changed (negative)," suggest that nurses who return to work, faced with the practical time and energy constraints of childcare, begin to prioritize values such as family stability and child-rearing over external rewards such as promotion and social recognition. In particular, the subcategory "My thoughts and actions at work have changed (negative)" included statements that directly expressed constraints on status and career advancement, such as *"It takes time to build a career with reduced hours"* (Participant 107). This suggests that the decline in prestige work values shown in the quantitative analysis is caused by structural factors such as the reallocation of roles and time due to childcare. Previous research reviewing how nurses with children balance childcare and their careers also found that nurses struggled with dual roles and compromised their career aspirations to prioritize their families, consistent with the results of this study [25].

Regarding extrinsic work values, the number of participants who showed increases (58) was the largest among the four factors. This suggests that the impact of childcare on household finances and a growing desire for a stable income, as indicated by the subcategory " My sense of money changed," may have promoted these values among participants. The maintenance of intrinsic work values suggests that nurses' professional identities and continued interest in their work remain core to their careers, even after changes in their roles due to childcare. Furthermore, in the subcategory "My thoughts and actions at work have changed" of the category "My thoughts and actions have changed," statements such as *"I have been able to think more carefully and empathize with things by imagining what would happen to my family if what is happening to the patient in front of me were happening to my family"* (Participant 116) indicated increased empathy and a new perspective on work. This suggests that professional growth (intrinsic work values) may be qualitatively deepening beyond quantitative increases or decreases in scores.

### Factors associated with changes in work values

GEE analysis quantitatively revealed that changes in work values are strongly influenced by individual life-stage variables and social support. The most stable predictor was age at first child, which showed a positive association with changes in all work values: intrinsic, extrinsic, social, and prestige. Despite a trend toward a decline in prestige work values, older age at first child indicates either a smaller decline or a greater increase in each value. This consistent positive association suggests that, after the most demanding period of early child-rearing passes, as children grow older (approaching elementary school entry), values are reinvigorated and interest in self-actualization and achievement at work is somewhat restored. Furthermore, baseline values scores before pregnancy were the strongest predictor of changes in all values, supporting previous findings showing a high degree of stability in the relative ranking and individuality of values [2,3].

Beyond these general trends, it is important to consider the specific predictors of each value domain. First, increased intrinsic work values were positively associated with higher levels of multidimensional social support from family, friends, and significant others. These results suggest that having extensive support outside of the workplace serves as an important buffer, allowing nurses to maintain focus on essential professional goals, such as "skill improvement" and "personal growth," even after a career interruption. This finding is consistent with previous studies highlighting the importance of family support as a key factor in career management and work-life balance for nurses raising children [23,25]. Furthermore, as indicated by the qualitative category "Family and work must be balanced," this finding supports the structural importance of work-life balance, indicating that support outside of work influences work values.

In contrast, increased extrinsic work values were associated with younger nurses and full-time employment. This suggests that younger nurses—with longer careers, increasing financial responsibilities, and full-time employment, which emphasizes stable income and employment—are strongly motivated to balance work and childcare. This result is consistent with the subcategories identified in the qualitative analysis, such as specific challenges like "I work hard for my family" and economic aspects like "My sense of money changed." The finding that the importance of stable income, which stems from the responsibilities of becoming a parent and raising a child (i.e.,) extrinsic work values, is associated with full-time employment is convincing.

Regarding social work values, increases were positively associated with a university/graduate school education, consistent with previous research showing that nurses with higher levels of education tend to be more specialized and have a stronger sense of social contribution [34]. This suggests that nurses may have reaffirmed and strengthened their values of contributing to society through their child-rearing experiences. However, many demographic factors and workplace characteristics, such as marital status, family structure, health status, whether they work night shifts, and working hours, did not show a statistically significant association with changes in work values across all four domains. This suggests that factors such as subjective perceived support and life stage (age of first child) have a stronger influence on the transformation of individual values than structural factors.

Furthermore, workplace social support was not found to be related to changes in work values in this study. Previous studies have also highlighted that nurses who are parents often lack organizational support from supervisors and other sources [25,35], suggesting that participants in this study may not have received support tailored to their needs. This lack of support may be reflective of broader systemic issues, such as the "motherhood penalty," which means working mothers are perceived as less committed to their work because they must balance work with childcare responsibilities. Therefore, they may miss out on career advancement opportunities [36]. In Japan, where the gender gap index is relatively low, female doctors with children earn 37.2% less than their male counterparts [37]. In Japan, where such a culture is deeply ingrained, nursing managers need to implement measures to address the gender gap challenge in nursing organizations with a predominantly female workforce.

## Policy and practice implications

The findings of this study provide important policy and practical implications for Japan's nursing shortage, which is intensified due to the aging population and barriers to women's career continuation. Nursing managers should understand that a decline in prestige work values represents a temporary change in priorities, not a permanent loss of promotional ambition among returning nurses. Therefore, rather than forcing nurses onto a full-time promotion path immediately after returning to work, introducing a "part-time career path" or "stepwise promotion model" that recognizes short-term and flexible work arrangements is important for preventing talented personnel from leaving the profession and enabling them to resume their careers in the future [24]. Moreover, support from home, the community, and within the workplace is essential for nurses to maintain their professionalism. Hospitals can indirectly support career continuation by providing nurses' families with information on childcare resources and conducting awareness-raising activities regarding the importance of sharing responsibilities within the home. Furthermore, as the age of the first child was a

stable predictor of changes in all values, interventions should gradually shift from focusing on flexible working hours immediately after returning to work (when children are young) to focusing on skill development and role expansion (intrinsic and social work values) as children grow older. The tendency for pregnancy and childbirth to shift priorities toward conservative values—rather than status or extrinsic success— is a key factor behind the global phenomenon of women's career interruptions [4], with direct implications for the retention of the international nursing workforce. Similar to Japan's experience, in other countries with large gender gaps [11], delayed institutional support may lead to the early redefinition of prestige work values, resulting in a decline in values. From an international perspective, we suggest that flexible working hours, access to high-quality public childcare services, and career support to prevent the "motherhood penalty" are key to global nurse retention strategies [36,37].

### Study limitations

This study has several limitations. First, its retrospective cross-sectional design may have introduced recall bias. Recall bias has been shown to be influenced by factors such as the time elapsed from the target period to the present, the abstractness and complexity of the topic, the importance and social desirability of the concept, and the participants' motivation level [16]. In particular, pre-pregnancy work values may be distorted by current circumstances and values. Specifically, participants might retrospectively overestimate their previous commitment to "prestige" values to contrast them with their current focus on family, or conversely, their current fatigue from childcare might skew their memories of their pre-pregnancy work life. However, major life events like pregnancy often serve as powerful "temporal anchors," which can enhance the accuracy of retrospective recall by providing a clear point of reference. While we attempted to mitigate this bias by setting a clear anchor point and using an established scale, acknowledging the potential for social desirability and memory reconstruction remains essential. Future prospective cohort studies that follow nurses on a larger scale may overcome these limitations.

Second, as the data were collected through an online survey, the sample was likely biased toward generations with easy internet access, thereby limiting the generalizability of the results to the entire Japanese nurse population. The survey can be completed on various devices, including smartphones and tablets, and differences in screen size and interface may have influenced survey participation and response behavior. As this survey was voluntary, the participant selection may have been biased toward those who were interested in the research topic. Data from people with a vested interest or opinion regarding the topics discussed in this study may have been overrepresented.

Third, although we incorporated supplementary qualitative data into a primarily quantitative design, the qualitative analysis was limited to open-ended responses and did not provide the deep insights that could be obtained from in-depth interviews. Therefore, it only served to support and contextualize the quantitative results.

Finally, the participants were limited to nurses working in Japan. Japan has a large gender gap [11]; thus, childcare and work balance may differ from other countries. The findings may not apply to other countries or cultural backgrounds. Therefore, similar studies should be conducted in different countries and cultural areas to achieve wider applicability.

### Conclusion

This study, using quantitative data and supplemental qualitative insights, revealed that prestige work values decline among Japanese nurses returning to work after pregnancy and childbirth. This decline is due to time constraints and structural changes in priorities caused by childcare. Furthermore, GEE analysis showed that the age of the first child and multidimensional social support from family, friends, and significant others play an important role in changing work values. These findings provide foundational evidence base for nursing managers to develop more flexible and individualized support strategies to support the careers of nurses raising children and prevent them from leaving the profession. Future longitudinal studies are needed to eliminate recall bias and further verify the causal relationships and long-term effects of changes in work values.

## Supporting information

**S1 Appendix. Questionnaire.** This is the S1 Appendix: Questionnaire.
(DOCX)

**S2 Appendix. Data for creating figures.** This is the S2 Appendix: Data for creating figures.
(XLSX)

## Acknowledgments

We want to thank all the nurses who participated in this study.

## Author contributions

**Conceptualization:** Yukari Hara, Aoi Nakagawa, Junko Omori.

**Data curation:** Yukari Hara, Aoi Nakagawa.

**Formal analysis:** Yukari Hara.

**Funding acquisition:** Yukari Hara, Junko Omori.

**Investigation:** Yukari Hara, Aoi Nakagawa.

**Methodology:** Yukari Hara, Aoi Nakagawa.

**Project administration:** Yukari Hara.

**Software:** Yukari Hara.

**Supervision:** Junko Omori.

**Writing – original draft:** Yukari Hara.

**Writing – review & editing:** Yukari Hara, Aoi Nakagawa, Junko Omori.

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
