## [Decision Letter · Decision Letter 0]

7 Nov 2025

Dear Dr. Hara,

Thank you for submitting your manuscript to PLOS ONE. After careful consideration, we feel that it has merit but does not fully meet PLOS ONE’s publication criteria as it currently stands. Therefore, we invite you to submit a revised version of the manuscript that addresses the points raised during the review process.

We look forward to receiving your revised manuscript.

Kind regards,

Paridhi Jha, PhD

Academic Editor

PLOS ONE

Journal Requirements:

3. Please expand the acronym “JSPS” (as indicated in your financial disclosure) so that it states the name of your funders in full.

This research was funded by JSPS KAKENHI (grant number 22K17434) and TUMUG Support Program from Center for Diversity, Equity, and Inclusion, Tohoku University.

5. We note that you have indicated that there are restrictions to data sharing for this study. For studies involving human research participant data or other sensitive data, we encourage authors to share de-identified or anonymized data. However, when data cannot be publicly shared for ethical reasons, we allow authors to make their data sets available upon request. For information on unacceptable data access restrictions, please see http://journals.plos.org/plosone/s/data-availability#loc-unacceptable-data-access-restrictions.

Additional Editor Comments:

Clarify and strengthen the theoretical framework surrounding work values and their transformation, ensuring that the conceptual basis is well-defined and supported by relevant literature.

Revise the methodology section to enhance transparency and rigor, particularly in the qualitative component. The procedures for data collection, coding, and analysis need to be described in greater detail.

Improve statistical reporting by including effect sizes, confidence intervals, and model diagnostics. This will allow for a more robust interpretation of the findings.

Enhance the integration of qualitative and quantitative findings to justify the use of a mixed-methods approach. Currently, the connection between both strands of data is weak and underdeveloped.

Provide access to anonymized data or offer a more detailed justification for the restrictions imposed. The current data availability statement does not meet PLOS ONE’s transparency standards.

Conduct a thorough language revision to improve clarity, grammar, and readability. While the manuscript is generally intelligible, several sections contain awkward phrasing and typographical errors.

These revisions are essential to ensure the manuscript meets the journal’s standards for methodological rigor, transparency, and clarity.

Reviewers' comments:

Reviewer's Responses to Questions

**Comments to the Author**

1. Is the manuscript technically sound, and do the data support the conclusions?

Reviewer #1: Yes

Reviewer #2: No

Reviewer #3: Yes

Reviewer #4: Yes

2. Has the statistical analysis been performed appropriately and rigorously?

Reviewer #1: Yes

Reviewer #2: No

Reviewer #3: Yes

Reviewer #4: Yes

3. Have the authors made all data underlying the findings in their manuscript fully available?

Reviewer #1: Yes

Reviewer #2: No

Reviewer #3: No

Reviewer #4: No

4. Is the manuscript presented in an intelligible fashion and written in standard English?

Reviewer #1: No

Reviewer #2: Yes

Reviewer #3: Yes

Reviewer #4: Yes

Reviewer #1: Thank you for the manuscript.

The manuscript is sound and very relevant in workforce development. Here are a few comments/recommendations for you to consider to strengthen the paper for publication.

Here are some of the strengths of the paper.

The use of a retrospective cross-sectional survey is very practical given the difficulty of longitudinal tracking in this population. The paired t-tests and Generalized Estimating Equations (GEE) are suitable for analyzing within-subject changes and identifying associated factors. Your study also employed the use of established scales for work values (Shortened Nurses' Work Values Scale), social support (MSPSS), and workplace support, which enhanced measurement reliability. Again, power analysis using GPower was clearly reported, with a sample size of 199 meeting the requirements for regression analysis with adequate power.

Below are some recommendations for you to address.

1. The title of the manuscript makes it look like it is a mixed-methods paper; however, the execution of the study relies heavily on quantitative methods/analysis. There is very limited or no integration of qualitative methods. Can you explain why this is the case?

There are no clear qualitative research questions or a rationale for a mixed methods approach.

Again, the qualitative analysis is described with no mention of coding procedures and analysis.

I recommend that you substantiate the mixed-methods claim by elaborating the qualitative design (sampling, coding, integration strategy) or reframing the study as a primarily quantitative design with supplemental qualitative insights.

You would have to justify the use of the mixed methods approach.

2. The quantitative findings of the study are significant; however, there is no clear demonstration as to how the qualitative findings reinforce or contrast with the quantitative results. The research topic indicates that the paper is a mixed-method, but there is insufficient qualitative information to back or justify this method.

3. On statistical analysis, the manuscript lacks detail on the correlation structure used in GEE and whether model diagnostics were performed. You would also need to justify the rationale for the effect size selected for the power analysis.

4. Please review the abstract for coherence and consistency with the findings of the paper. You may consider being brief with respect to the methodology and reducing the word count.

5. The discussion could be expanded to consider implications for nursing policy and workforce retention strategies in Japan, especially in light of the aging population and gender equity challenges.

6. Consider including a limitations subsection that explicitly addresses recall bias, generalizability, and the cross-sectional design.

Reviewer #2: Dear Authors,

Thank you for your submission. Your study addresses an important and underexplored topic: the transformation of work values among Japanese nurses before and after pregnancy. However, several critical issues must be addressed to improve the manuscript’s scientific rigor, clarity, and overall contribution. Below are detailed comments and suggestions organized by section:

Abstract

• The abstract is overly dense and includes methodological jargon that may not be accessible to all readers. Consider simplifying and clarifying the structure.

• The phrase “pioneering mixed-methods study” is overstated given the limitations of the design and analysis.

• The conclusion should better reflect the limitations and avoid generalizations not supported by the data.

Introduction

• The literature review is superficial. Key concepts such as “work values” and “value transformation” are not sufficiently defined or contextualized.

• The rationale for focusing on Japanese nurses is clear, but the global relevance of the findings is overstated.

• The introduction would benefit from a clearer articulation of the research questions and hypotheses.

Methods

• The retrospective design is problematic. The authors acknowledge this but do not provide sufficient mitigation strategies.

• The use of G*Power to justify sample size is appropriate, but the choice of effect size (0.15) and power (0.95) should be justified in the context of prior literature.

• The description of the survey instrument lacks detail. How were the questions validated? Were they piloted?

• The qualitative component is inadequately described. What was the coding process? How were themes derived? Was intercoder reliability assessed?

Statistical Analysis

• The use of paired t-tests is simplistic and does not account for potential confounders.

• The application of Generalized Estimating Equations (GEE) is appropriate but poorly explained. What correlation structure was used? Were model diagnostics performed?

• The results section lacks effect sizes and confidence intervals, which are essential for interpreting the magnitude and precision of findings.

• There is no discussion of missing data or how it was handled.

Results

• The presentation of results is fragmented. Tables are referenced but not included in the review document.

• The finding that only prestige work values decreased significantly is interesting but not explored in depth.

• The GEE results are presented without sufficient interpretation. What do these associations mean in practical terms?

Qualitative Analysis

• The qualitative findings are superficial. There is no evidence of methodological rigor (e.g., thematic saturation, triangulation).

• Quotes from participants are not included, which limits the reader’s ability to assess the validity of the interpretations.

• The integration of qualitative and quantitative findings is weak. Consider using a joint display or narrative synthesis.

Discussion

• The discussion overstates the implications of the findings. The limitations of the design and analysis should be more prominently acknowledged.

• The authors should avoid normative statements such as “Understanding these shifts is crucial…” unless supported by evidence.

• The discussion lacks engagement with broader literature on gender, work-life balance, and nursing retention.

Ethics and Data Availability

• The ethics statement is thorough and appropriate.

• However, the data availability is insufficient. PLOS ONE requires full data transparency. The authors must provide anonymized datasets or justify restrictions more clearly.

Language and Style

• The manuscript is generally intelligible but contains awkward phrasing and grammatical inconsistencies. A professional language edit is recommended.

• Examples: “not very import” (likely a typo), “prestige work values showed a significant decrease in the mean” (awkward phrasing).

Recommendations for Improvement

1. Clarify and strengthen the theoretical framework around work values and their transformation.

2. Revise the methodology section to provide greater transparency and rigor, especially for the qualitative component.

3. Improve statistical reporting by including effect sizes, confidence intervals, and model diagnostics.

4. Enhance the integration of qualitative and quantitative findings to justify the mixed-methods approach.

5. Provide access to anonymized data or a detailed justification for restrictions.

6. Conduct a thorough language revision to improve clarity and readability.

Reviewer #3: ABSTRACT

• In the abstract, the results section speaks only to the quantitative findings. Briefly mention at least one key qualitative insight from the results.

BACKGROUND

• Line 64- The sentence starting at line 64 is virtually a repetition of the sentence starting at line 49. Consider combining the two paragraphs or improving their flow.

METHODS

• Line 180- keep the same referencing style

• Lines 195 to 197- How qualitative data was collected is not clear. Authors should indicate the data collection method used, whether by face-to-face interview or focus group discussion. Which tool was used to collect the data? Was it by audio recording or text?

• The qualitative aspect of data collection lacks rigor. It does not address questions such as “what type of data analysis was used? Was it content or thematic analysis?”, “Who did the coding?”, “How was the coding done?” and “Whether any software was used in managing the data?” among others. Authors should rectify this.

• Line 90- The Authors mentioned Retrospective recall bias as part of the limitations. Authors should go beyond citing Thigpen and Hipp to indicate in the methods section, steps taken to mitigate this, to improve the validity of the study approach.

• Line 98- In Japan, preschool covers a wide age range; therefore, specifying the age range of their first child will be more precise.

• Variables were well listed; however, to improve readability, I suggest the authors use a table either in the manuscript or on a different sheet as a supplementary document to capture the variables listed.

RESULTS

• Line 275- insert the necessary figures or remove the phrase.

DISCUSSION

• Either by Omission or misunderstanding on my part, I did not find an explanation from the qualitative data suggesting why there is a drop in prestige. Authors should indicate if this was just a feeling of the participants or otherwise.

Reviewer #4: This manuscript addresses an important and underexplored topic. However, some revisions are needed to improve clarity, structure, and analytical depth.

Study Design and Recall Bias: The use of a retrospective cross-sectional design is acknowledged, but here recall bias remains a concern. Authors should further justify the reliability of recalling pre-pregnancy work values and discuss how this may have influenced results.

Methodological Detail: Please clarify the correlation structure assumed in the GEE model

Sample Representation: The sampling process is not fully explained. How samples were selected and how was the web survey done?

Ethics Section: Clarify whether consent was implied or explicitly obtained through digital checkboxes.

Qualitative Analysis: The qualitative analysis section needs clearer explanation of the coding process (e.g., number of coders, inter-rater agreement, any use of software).

The inclusion of participant IDs in Table 5 is appreciated. If feasible, consider adding a brief note in the methods to explain that these IDs

Integration of Mixed Methods: The mixed-methods design would be stronger with explicit integration—how do the qualitative findings confirm, complement, or contradict quantitative results?

Data Availability: PLOS ONE requires publicly available data. Data availability statement is incomplete and non-compliant with PLOS ONE policy.

Discussion: Expand on implications for hospital policy and management beyond Japan.

References: DOI links should be active and complete.

.

Reviewer #1: No

Reviewer #2: **Yes:** RICHARDSON AUGUSTO ROSENDO DA SILVARICHARDSON AUGUSTO ROSENDO DA SILVARICHARDSON AUGUSTO ROSENDO DA SILVARICHARDSON AUGUSTO ROSENDO DA SILVA

Reviewer #3: **Yes:** Clement Asoya AsebigaClement Asoya AsebigaClement Asoya AsebigaClement Asoya Asebiga

Reviewer #4: No

---

## [Author Response · Author response to Decision Letter 1]

23 Dec 2025

Responses to Reviewers’ Comments

Paper Title: An exploratory analysis of changes in work values among nurses before and after pregnancy

Journal Name: PLOS ONE

We would like to express our sincere gratitude for your review of our paper and your valuable and constructive feedback. All comments have been extremely helpful in improving the quality and transparency of this study, and we have thoroughly revised the manuscript based on your suggestions. Our responses to the comments are provided below. The revised sections are highlighted in red within the manuscript.

Editor Comments

Comment 1: Clarify and strengthen the theoretical framework surrounding work values and their transformation, ensuring that the conceptual basis is well-defined and supported by relevant literature.

Response 1:

Thank you for your valuable comment. We have significantly revised the Introduction and Discussion sections to strengthen the theoretical background regarding work values and their transformation. Based on previous research, we have enriched the discussion of the impact of life events (particularly pregnancy and childcare) on career values, clarifying the conceptual foundation of this study.

Revisions: Introduction and Discussion sections

Comment 2: Revise the methodology section to enhance transparency and rigor, particularly in the qualitative component. The procedures for data collection, coding, and analysis need to be described in greater detail.

Response 2:

Thank you for your advice. We have revised the entire Methods section and added more details, especially about the procedures for collecting and analyzing qualitative data. This includes the method for collecting qualitative data (open-ended questions), the number of analysts, the process for deriving categories and subcategories, and the fact that no software was used for analysis.

Revision: Methods

Comment 3: Improve statistical reporting by including effect sizes, confidence intervals, and model diagnostics. This will allow for a more robust interpretation of the findings.

Response 3:

Based on your comment, we have added effect sizes (Cohen's d) and 95% confidence intervals to the statistical analysis results. We have also added information about GEE model diagnostics to the methodology to strengthen the robustness of the interpretation of the results.

Revision: Methods/Results sections

Comment 4: Enhance the integration of qualitative and quantitative findings to justify the use of a mixed-methods approach. Currently, the connection between both strands of data is weak and underdeveloped.

Response 4:

Reviewers #1 and #4 made similar comments, so we reframed this study as "primarily a quantitative design with supplemental qualitative insights" and significantly strengthened the integration of qualitative and quantitative findings in the discussion.

Revision: Discussion section

Comment 5: Provide access to anonymized data or offer a more detailed justification for the restrictions imposed. The current data availability statement does not meet PLOS ONE’s transparency standards.

Response 5:

We have revised our data availability statement to comply with PLOS ONE policy. Specifically, we now state, "All relevant data are within the manuscript and its Supporting Information files."

Comment 6: Conduct a thorough language revision to improve clarity, grammar, and readability. While the manuscript is generally intelligible, several sections contain awkward phrasing and typographical errors.

Response 6:

Following your suggestion, the language of the manuscript was reviewed and enhanced by a professional English editing service. Further, we focused on correcting grammatical errors, ensuring clarity of expression, refining the academic tone, and improving the overall fluency and readability of the text. ________________________________________

Reviewer #1 Comments

Comment 1: The use of a retrospective cross-sectional survey is very practical given the difficulty of longitudinal tracking in this population. The paired t-tests and Generalized Estimating Equations (GEE) are suitable for analyzing within-subject changes and identifying associated factors. Your study also employed the use of established scales for work values (Shortened Nurses' Work Values Scale), social support (MSPSS), and workplace support, which enhanced measurement reliability. Again, power analysis using GPower was clearly reported, with a sample size of 199 meeting the requirements for regression analysis with adequate power.

Response 1:

Thank you very much for your understanding of the strengths and characteristics of this study. Your insightful suggestions have significantly improved this manuscript.

Comment 2: The title of the manuscript makes it look like it is a mixed-methods paper; however, the execution of the study relies heavily on quantitative methods/analysis. There is very limited or no integration of qualitative methods. Can you explain why this is the case? There are no clear qualitative research questions or a rationale for a mixed methods approach. Again, the qualitative analysis is described with no mention of coding procedures and analysis. I recommend that you substantiate the mixed-methods claim by elaborating the qualitative design (sampling, coding, integration strategy) or reframing the study as a primarily quantitative design with supplemental qualitative insights. You would have to justify the use of the mixed methods approach.

Response 2:

Thank you for your valuable comments. We deeply regret that the mixed methods approach of this study was unclear.

1. Changes to the research design: Based on your comments, we restructured this study as a "primarily quantitative design with supplemental qualitative insights" and clarified the reasons for its adoption in the Introduction and Methods sections.

2. Clarification of the qualitative research objectives: We clearly stated that the qualitative questions (open-ended text) intended to delve deeper and contextualize the quantitative results.

3. Strengthening the methodology: We have enhanced transparency by providing detailed descriptions of the qualitative data sampling, analysis procedures, and integration strategy with the quantitative results.

Revisions: Introduction, lines 81-84 / Methods, lines 88-90, 273-289

Comment 3: The quantitative findings of the study are significant; however, there is no clear demonstration as to how the qualitative findings reinforce or contrast with the quantitative results. The research topic indicates that the paper is a mixed-method, but there is insufficient qualitative information to back or justify this method.

Response 3:

Thank you for your comment. In the discussion section, we specifically integrated how the qualitative findings supported the quantitative results (e.g., a decline in prestige work values) or contextualized the underlying personal and professional conflicts. This clearly demonstrated the value of using qualitative findings.

Revisions: The entire Discussion section

Comment 4: On statistical analysis, the manuscript lacks detail on the correlation structure used in GEE and whether model diagnostics were performed. You would also need to justify the rationale for the effect size selected for the power analysis.

Response 4:

Based on your comments, we have revised the Methods section.

1. GEE details: We have added information about the exchangeable structure used in the GEE analysis and the model diagnostics (QIC comparison) that supported the adoption of that structure.

2. Rationale for power analysis: We have clarified that the effect size (0.15) used in the power analysis was selected based on previous research, and have justified this rationale.

Revisions: Methods, lines 263-269 and 119-120

Comment 5: Please review the abstract for coherence and consistency with the findings of the paper. You may consider being brief with respect to the methodology and reducing the word count.

Response 5:

Based on your comments, we have revised the abstract. We have removed redundant technical terms and simplified the methodology description. Additionally, we have ensured consistency with the findings of the paper by briefly including key quantitative findings and supporting important qualitative insights.

Revisions: Throughout the Abstract

Comment 6: The discussion could be expanded to consider implications for nursing policy and workforce retention strategies in Japan, especially in light of the aging population and gender equity challenges.

Response 6:

Thank you for your suggestion. We have added a new subsection, "Policy and Practice Implications," to the Discussion section to delve deeper into the policy and practice implications that changes in work values have on nurse retention and career support in the context of Japan's aging population and the promotion of gender equality.

Revision: Discussion, lines 511-535

Comment 7: Consider including a limitations subsection that explicitly addresses recall bias, generalizability, and the cross-sectional design.

Response 7:

Based on your comments, we have significantly revised the "Study limitations" subsection at the end of the discussion. We now clearly discuss the recall bias resulting from the retrospective cross-sectional design, the limitations on generalizability resulting from web-based sampling, and how to address these issues.

Revision: Study limitations, lines 537-561

Reviewer #2 Comments

Dear Authors,

Thank you for your submission. Your study addresses an important and underexplored topic: the transformation of work values among Japanese nurses before and after pregnancy. However, several critical issues must be addressed to improve the manuscript’s scientific rigor, clarity, and overall contribution. Below are detailed comments and suggestions organized by section:

Response:

Thank you very much for understanding the strengths of this study. Your advice has helped us further improve this manuscript.

Abstract

Comment: The abstract is overly dense and includes methodological jargon that may not be accessible to all readers. Consider simplifying and clarifying the structure. The phrase “pioneering mixed-methods study” is overstated given the limitations of the design and analysis. The conclusion should better reflect the limitations and avoid generalizations not supported by the data.

Response:

Based on your comments, we have made significant revisions to the abstract. Specifically, we have avoided excessive methodological jargon and written it succinctly, focusing on the core of the study and its main results. Furthermore, we have removed the overly explicit phrase "pioneering mixed-methods study." We have also revised the conclusion to be more cautious and evidence-based, presenting the main findings while keeping in mind the limitations of the design.

Introduction

Comment: The literature review is superficial. Key concepts such as “work values” and “value transformation” are not sufficiently defined or contextualized. The rationale for focusing on Japanese nurses is clear, but the global relevance of the findings is overstated. The introduction would benefit from a clearer articulation of the research questions and hypotheses.

Response:

Based on your comments, we have revised the Introduction to improve its logical flow and eliminate redundancies.

1. Conceptual clarification: We conducted a thorough literature review of "work values" and "changes in values," establishing clear definitions and a solid theoretical foundation of these key concepts.

2. Global relevance: While focusing the discussion on the context of Japanese nurses, we adjusted the format to discuss universal implications of the results (e.g., issues related to balancing work and family life) and avoided overgeneralization.

3. Research questions and hypotheses: We clearly stated the quantitative research questions. However, as you acknowledge, this research area—exploring changes in work values among mothers from pre-pregnancy to post-natal care after childbirth and childcare—including the factors influencing these changes, is still unexplored. As this study was positioned as an exploratory analysis, it was difficult to formulate clear hypotheses.

Revisions: The entire Introduction section

Methods

Comment: The retrospective design is problematic. The authors acknowledge this but do not provide sufficient mitigation strategies. The use of G*Power to justify sample size is appropriate, but the choice of effect size (0.15) and power (0.95) should be justified in the context of prior literature. The description of the survey instrument lacks detail. How were the questions validated? Were they piloted? The qualitative component is inadequately described. What was the coding process? How were themes derived? Was intercoder reliability assessed?

Response:

The methods section was thoroughly revised.

1. Recall bias: In addition to the study limitations section, measures to enhance the reliability of retrospective data (e.g., clarity of questions, facilitation of recall based on specific life events) were added to the Methods section. (Lines 100-109)

2. Power rationale: The selection of effect size (0.15) and power (0.95) were justified based on previous research. (Lines 119-120)

3. Questionnaire description: For scales (e.g., Shortened Nurses' Work Values Scale, MSPSS), the original development papers were cited and descriptions of the scale structure, subscales, and reliability (Cronbach's alpha) used in this study were added. Since these scales have been validated in previous studies, no pilot studies or revalidation was conducted in this study. Furthermore, the variables described in the Methods section were added to the supplementary material (S1 Appendix) to improve readability and clarify how they were used in the questionnaire.

4. Qualitative data analysis: We have strengthened the transparency of our qualitative data analysis by providing detailed descriptions of the analytical procedures and methods, the number of analysts, and the procedures for verifying the validity of the analysis among researchers. (Lines 273-289)

Analysis of quantitative variables

Comment: The use of paired t-tests is simplistic and does not account for potential confounders. The application of Generalized Estimating Equations (GEE) is appropriate but poorly explained. What correlation structure was used? Were model diagnostics performed? The results section lacks effect sizes and confidence intervals, which are essential for interpreting the magnitude and precision of findings. There is no discussion of missing data or how it was handled.

The presentation of results is fragmented. Tables are referenced but not included in the review document. The finding that only prestige work values decreased significantly is interesting but not explored in depth. The GEE results are presented without sufficient interpretation. What do these associations mean in practical terms?

Response:

The statistical analysis, results, and discussion sections have been significantly revised.

1. Role of t-tests and GEE: We clarified that t-tests were used to demonstrate simple descriptive changes, and that GEE was the primary analysis, including potential confounders. (Lines 227-232, 244-246)

2. Details and reporting of t-test and GEE results: We added details on correlation structures and model diagnostics. Additionally, we added effect sizes and 95% confidence intervals to all statistical results. Furthermore, we provided additional textual explanations for significant results presented in tables. (Results section)

3. Handling of missing values: There were no missing values for any variables, and this was clearly stated. (Lines 242-243)

4. Interpretation of results: We integrated the "decrease in prestige work values" and other results with qualitative findings (e.g., work-life balance, time constraints) and added an interpretation in the discussion section that delves deeper into their practical implications (e.g., decreased interest in promotion). (Results and Discussion sections)

Qualitative descriptive analysis of descriptive statistics

Comment: The qualitative findings

---

## [Decision Letter · Decision Letter 1]

20 Jan 2026

Dear Dr. Hara,

Thank you for submitting your manuscript to PLOS ONE. After careful consideration, we feel that it has merit but does not fully meet PLOS ONE’s publication criteria as it currently stands. Therefore, we invite you to submit a revised version of the manuscript that addresses the points raised during the review process.

We look forward to receiving your revised manuscript.

Kind regards,

Xiangdan Piao

Academic Editor

PLOS One

Journal Requirements:

Reviewers' comments:

Reviewer's Responses to Questions

**Comments to the Author**

Reviewer #1: All comments have been addressed

Reviewer #3: All comments have been addressed

Reviewer #4: All comments have been addressed

2. Is the manuscript technically sound, and do the data support the conclusions?

Reviewer #1: Yes

Reviewer #3: Yes

Reviewer #4: Yes

3. Has the statistical analysis been performed appropriately and rigorously?

Reviewer #1: Yes

Reviewer #3: Yes

Reviewer #4: Yes

4. Have the authors made all data underlying the findings in their manuscript fully available?

Reviewer #1: Yes

Reviewer #3: Yes

Reviewer #4: Yes

5. Is the manuscript presented in an intelligible fashion and written in standard English?

Reviewer #1: Yes

Reviewer #3: Yes

Reviewer #4: Yes

Reviewer #1: Congratulations on making significant improvements to the manuscript and responding to all reviewer and editor comments. This has made the manuscript very strong, and it now has a clearer theoretical grounding of work values and value transformation. There is equally a much improved qualitative analysis section. the results section and the discussion section are well written and coherent.

However, here are a few suggestions

1. in your abstract, as your keyword, you use the word/phrase "nursing stuff." It is not clear if that is what you really want to write. The actual word should be "nursing staff." What do you think? Please clarify and correct.

2. Your transitions from paragraph to paragraph also appear to be abrupt. Please double-check and ensure that there is a flow between the paragraphs and that they transition smoothly.

3. In the qualitative analysis, it will be appropriate to put the supporting statements in quotation marks (i.e., "...") and italicize them, alongside the participant code or pseudonym.

4. some small language copyediting of the entire manuscript will do.

Thank you

Reviewer #3: REVIEW COMMENTS

The authors have addressed almost all the comments raised previously. however, the authors need to address a few minor issues in relation to their ethical consideration and methodology.

ETHICAL CONSIDERATION

1. LINE 147; The last sentence states that the study does not include minors. That was very clear; however, the authors could briefly state why minors are not included to avoid queries (e.g., "As all participants were required to be nurses and parents, no minors were eligible for inclusion.").

ANALYSIS METHOD

1. I suggest the authors consider revising the heading "Analysis method" to "Data Analysis" as a top-level subsection. This sounds more standard.

2. In the qualitative part, the authors mention that "validity was assessed by two other researchers." That was perfect; however, I suggest they use the standard qualitative term "trustworthiness" or "rigor" in their description and specify the process. This strengthens the claim.

3. The author's description ("inductively grouped... abstract subcategories and categories…") is essentially ‘inductive thematic analysis.’ I therefore suggest that the authors be clear and use the term inductive thematic analysis. This term adds methodological recognition to their work.

Reviewer #4: The manuscript now largely meets the required standards. Only minor wording refinements are needed to avoid unintended longitudinal implications, along with a brief clarification on the direction of recall bias. No major methodological issues remain.

.

Reviewer #1: No

Reviewer #3: **Yes:** Clement Asoya AsebigaClement Asoya AsebigaClement Asoya AsebigaClement Asoya Asebiga

Reviewer #4: **Yes:** Amisha S AminAmisha S AminAmisha S AminAmisha S Amin

---

## [Author Response · Author response to Decision Letter 2]

8 Feb 2026

Responses to Reviewers’ Comments

Paper Title: An exploratory analysis of changes in work values among nurses before and after pregnancy

Journal Name: PLOS ONE

We would like to express our sincere gratitude for your further review of our paper and for your valuable and constructive feedback. We are encouraged by the positive assessment of our revisions. We have addressed the remaining points as suggested, which we believe have further polished the manuscript. Our responses to the comments are provided below. The revised sections are highlighted in red in the manuscript.

Reviewer #1 Comments

Comment: Congratulations on making significant improvements to the manuscript and responding to all reviewer and editor comments. This has made the manuscript very strong, and it now has a clearer theoretical grounding of work values and value transformation. There is equally a much improved qualitative analysis section. the results section and the discussion section are well written and coherent.

Response: Thank you very much for your kind words and for recognizing the improvements made to our manuscript. We truly appreciate the time and effort you have dedicated to providing such constructive feedback throughout the review process.

Comment: However, here are a few suggestions

1. in your abstract, as your keyword, you use the word/phrase "nursing stuff." It is not clear if that is what you really want to write. The actual word should be "nursing staff." What do you think? Please clarify and correct.

Response 1: Thank you very much for pointing out this typographical error. We have corrected "nursing stuff" to "nursing staff" in the keywords section of the abstract. We apologize for the oversight.

Revisions: Abstract (Keywords)

Comment: 2. Your transitions from paragraph to paragraph also appear to be abrupt. Please double-check and ensure that there is a flow between the paragraphs and that they transition smoothly.

Response 2: We appreciate this suggestion. We have reviewed the entire manuscript and added transitional phrases or adjusted the opening sentences of the paragraphs to ensure a smoother logical flow and better connectivity between sections, particularly in the Introduction and Discussion.

Revisions: Throughout the manuscript.

Comment: 3. In the qualitative analysis, it will be appropriate to put the supporting statements in quotation marks (i.e., "...") and italicize them, alongside the participant code or pseudonym.

Response 3: Thank you for this specific guidance on formatting. We have revised the qualitative findings section so that all supporting qualitative statements (quotes) are now enclosed in quotation marks and italicized. We have also ensured that participant codes are consistently placed alongside each quote.

Revisions: Results (Qualitative findings), Discussion sections.

Comment: 4. some small language copyediting of the entire manuscript will do.

Response 4: We appreciate the suggestion. The entire manuscript has undergone further language copyediting to improve clarity, grammar, and overall readability.

Revisions: Throughout the manuscript.

Reviewer #3:

Comment: [REVIEW COMMENTS] The authors have addressed almost all the comments raised previously. however, the authors need to address a few minor issues in relation to their ethical consideration and methodology.

Response: Thank you for your valuable feedback. In response to your suggestion, we have further refined the Ethical Considerations and Methodology sections to ensure greater clarity and transparency.

Comment: [ETHICAL CONSIDERATION] 1. LINE 147; The last sentence states that the study does not include minors. That was very clear; however, the authors could briefly state why minors are not included to avoid queries (e.g., "As all participants were required to be nurses and parents, no minors were eligible for inclusion.").

Response: Thank you for this helpful suggestion. To clarify, we have added the following sentence to the "Ethical Considerations" section: "As all participants were required to be licensed nurses and parents, no minors were eligible for inclusion."

Revisions: Ethical considerations subsection in the Methods section.

Comment: [ANALYSIS METHOD] 1. I suggest the authors consider revising the heading "Analysis method" to "Data Analysis" as a top-level subsection. This sounds more standard.

Response: We agree that "Data Analysis" is more standard. We have updated the heading accordingly.

Revisions: Methods section (subheading).

Comment: [ANALYSIS METHOD] 2. In the qualitative part, the authors mention that "validity was assessed by two other researchers." That was perfect; however, I suggest they use the standard qualitative term "trustworthiness" or "rigor" in their description and specify the process. This strengthens the claim.

Response: Thank you for this suggestion. We used the terms "trustworthiness" and "rigor" instead of "validity" and provided a detailed explanation of the peer debriefing process to demonstrate the rigor of the qualitative analysis.

Revisions: Data Analysis subsection in the Methods section

Comment: [ANALYSIS METHOD] 3. The author's description ("inductively grouped... abstract subcategories and categories…") is essentially ‘inductive thematic analysis.’ I therefore suggest that the authors be clear and use the term inductive thematic analysis. This term adds methodological recognition to their work.

Response: We have revised the text to explicitly state that we employed "inductive thematic analysis." We agree that using this established term provides better methodological recognition for our qualitative approach.

Revisions: Throughout the manuscript.

Reviewer #4:

Comment: The manuscript now largely meets the required standards. Only minor wording refinements are needed to avoid unintended longitudinal implications, along with a brief clarification on the direction of recall bias. No major methodological issues remain.

Response: We are grateful for your positive assessment. We have addressed your remaining points as follows:

1. Wording Refinements: To avoid any implication of a prospective longitudinal design, we have consistently used terms such as "retrospective assessment" or "perceived changes" throughout the manuscript to clarify the cross-sectional nature of our data collection.

2. Recall Bias: We have added a discussion in the "Limitations" section regarding recall bias. We noted that while retrospective reports can be influenced by current circumstances, major life events like pregnancy often serve as "temporal anchors" that may improve recall accuracy, while acknowledging that social desirability bias could still play a role.

Revisions: Methods, Discussion (Limitations) sections.

Once again, we would like to thank you for your valuable comments and guidance. We believe that these revisions have made the manuscript stronger and clearer, meeting PLOS ONE's publication standards.

---

## [Decision Letter · Decision Letter 2]

7 Apr 2026

An exploratory analysis of changes in work values among nurses before and after pregnancy

PONE-D-25-37669R2

Dear Dr. Hara,

We’re pleased to inform you that your manuscript has been judged scientifically suitable for publication and will be formally accepted for publication once it meets all outstanding technical requirements.

Kind regards,

Xiangdan Piao

Academic Editor

PLOS One

Additional Editor Comments (optional):

Reviewers' comments:

Reviewer's Responses to Questions

**Comments to the Author**

Reviewer #1: All comments have been addressed

Reviewer #3: All comments have been addressed

Reviewer #4: All comments have been addressed

2. Is the manuscript technically sound, and do the data support the conclusions?

Reviewer #1: Yes

Reviewer #3: Yes

Reviewer #4: Yes

3. Has the statistical analysis been performed appropriately and rigorously?

Reviewer #1: Yes

Reviewer #3: Yes

Reviewer #4: Yes

4. Have the authors made all data underlying the findings in their manuscript fully available?

Reviewer #1: Yes

Reviewer #3: Yes

Reviewer #4: Yes

5. Is the manuscript presented in an intelligible fashion and written in standard English?

Reviewer #1: Yes

Reviewer #3: Yes

Reviewer #4: Yes

Reviewer #1: (No Response)

Reviewer #3: authors responded all issues raised

Reviewer #4: The manuscript is now technically a sound piece of scientific research with data that supports the conclusions. All comments addressed by authors

.

Reviewer #1: No

Reviewer #3: **Yes:** Clememt Asoya AsebigaClememt Asoya AsebigaClememt Asoya AsebigaClememt Asoya Asebiga

Reviewer #4: **Yes:** Amisha S AminAmisha S AminAmisha S AminAmisha S Amin

---

## [Editor Report · Acceptance letter]

PONE-D-25-37669R2

PLOS One

Dear Dr. Hara,

I'm pleased to inform you that your manuscript has been deemed suitable for publication in PLOS One. Congratulations! Your manuscript is now being handed over to our production team.

Kind regards,

on behalf of

Dr. Xiangdan Piao

Academic Editor

PLOS One